

# Scattering matrices of mineral dust aerosols: a refinement of the refractive index impact

Yifan Huang[1,2], Chao Liu[1,2,*], Yan Yin[1,2], Lei Bi[3]

[1]Collaborative Innovation Center on Forecast and Evaluation of Meteorological Disasters, Nanjing University of Information Science & Technology, Nanjing 210044, China
[2]Key Laboratory for Aerosol-Cloud-Precipitation of China Meteorological Administration, School of Atmospheric Physics, Nanjing University of Information Science & Technology, Nanjing 210044, China
[3]Department of Atmospheric Sciences, Zhejiang University, Hangzhou 310027, China

*Correspondence to*: Chao Liu (chao_liu@nuist.edu.cn)

**Abstract.** Dust, as one of the most important aerosols, plays a crucial role in the atmosphere by directly interacting with radiation, while there are significant uncertainties in determining dust optical properties to quantify radiative effects and to retrieve their properties. Both laboratory and in situ measurements show variations in dust refractive indices (*RI*s), and different *RI*s have been applied in different numerical studies of model developments, aerosol retrievals, and radiative forcing simulations. This study reveals the importance of the dust *RI* for the model development of its optical properties. The Koch-fractal polyhedron is used as the modeled geometry, and the pseudo-spectral time domain method and improved geometric-optics method are combined to cover optical property simulations over the complete size range. We find that the scattering matrix elements of different kinds of dust particles are reasonably reproduced by choosing appropriate *RI*s even using a fixed particle geometry. The uncertainty of the *RI* would greatly affect the determination of the geometric model, as a change in the *RI*, even in the widely accepted *RI* range, strongly affects the appropriate shape parameters to reproduce the measured dust phase matrix elements. A further comparison shows that the *RI* influences the scattering matrix elements differently from geometric factors, and, more specifically, the $P_{11}$, $P_{12}$, and $P_{22}$ elements seem more sensitive to dust *RI*. In summary, more efforts should be devoted to account for the uncertainties on the dust *RI* in modeling its optical properties, and the development of corresponding optical models can potentially be simplified by considering only variations over different *RI*s. Considerably more research, especially from direct measurements, should be carried out to better constrain the uncertainties related to the dust aerosol *RI*s.



# 1 Introduction

Atmospheric aerosols play an important role in the global radiation balance directly by scattering and absorbing incident solar radiation and indirectly by influencing cloud formation as CCN or IN (Chýlek et al., 1978; Sokolik et al., 2001; Yi et al., 2011). As a major type of aerosol, mineral dust is widely distributed in the globe, especially in arid regions. Mineral dust single-scattering properties, such as the single-scattering albedo and asymmetry factor, are fundamental not only to quantify their radiative effects, but also to develop satellite retrieval algorithms from optical observations (Kahn et al., 2005; Huang et al., 2014; Xu et al., 2017).

Several aerosol optical property databases, e.g., the Global Aerosol Data Set (GADS; Koepke et al., 1997) and the Optical Properties of Aerosols and Clouds (OPAC; Hess et al., 1998), have been built to meet the needs of aerosol remote sensing and radiation studies for all aerosols as well as for particular kinds of aerosols (Meng et al., 2010; Bi et al., 2018; Liu et al., 2019). However, for simplification, the optical properties of atmospheric aerosols are often investigated by assuming a relatively simple model, e.g., using spheres and spheroids, and this simplification has resulted in obvious errors (Mishchenko et al., 1997; Feng et al., 2009). For example, the measured phase functions of dust particles are distinctive compared with the phase functions of spherical particles at sideward and backward scattering angles (Koepke and Hess, 1988; Dobuvik et al., 2002; Nousiainen, 2009). Databases with better accuracy in characterizing dust optical properties can contribute to many aspects, such as radiative forcing calculations and remote sensing applications.

Several efforts have been devoted to studying the optical properties of dust aerosols (Bi et al., 2010; Meng et al., 2010; Ishimoto et al., 2010; Liu et al., 2013; Jin et al., 2016; Xu et al., 2017). Although it is difficult but possible to mathematically define the exact shape of an actual dust particle in numerical studies (Kahnert et al., 2014; Lindqvist et al., 2014), the use of a simplified but optically equivalent model is more convenient and doable (Nousiainen and Kandler, 2015; Liu et al., 2013). By using the measured particle size information and the assumed refractive index ($RI$), most studies treat the geometry as an unknown variable and look for proper geometric parameters that result in simulated optical properties consistent with measurements (Bi et al., 2010; Dubovik et al., 2006; Liu et al., 2013; Lin et al., 2018; Mishchenko et al., 1997; Osborne et al., 2011). Different nonspherical shapes have been developed and applied, such as a spheroid (Mishchenko et al., 1997; Dobuvik et al., 2002; Ge et al., 2011; Merikallio et al., 2011), an ellipsoid (Bi et al., 2009; Meng et al., 2010; Kemppinen et





al., 2015), and a superellipsoid (Bi et al., 2018b). Additionally, more complex and irregular particles have also been considered, e.g., spatial Poisson–Voronoi tessellation (Ishimoto et al., 2010), Gaussian random field (GRF) particles (Grynko et al., 2013), Koch-fractal particles (Liu et al., 2013; Jin et al., 2016), and nonsymmetric hexahedra (Bi et al., 2010; Liu et al., 2014). These "irregular" geometries as well as spheroids can achieve close agreement with measurements by using

appropriate shape parameters or combining the results from multiple shapes, which indicates that certain geometries may be optically similar or equivalent with respect to scattering light. Nousiainen and Kandler (2015) found that scattering properties of a cube-like dust particle can be mimicked by those of spheroids with a suitable shape distribution. Liu et al. (2014) found that the surface roughness and irregularity also share an optical equivalence. Lin et al. (2018) revealed that the scattering matrix elements of different types of dust particles can be achieved by changing only two parameters to specify

the geometry of the superspheroids. Such "optical equivalences" are important for practical applications using dust optical properties because we can use those from a relatively simple numerical model, instead those based on the exact and actual dust particles, for downstream remote sensing and radiative transfer applications.

In addition to the particle geometry, there are also significant uncertainties related to the dust $RI$, which is often considered an inherent characteristic (Kahnert and Nousiainen, 2006; Stegmann and Yang, 2017), whereas much less

attention has been devoted during the model development. Previous studies often assumed a fixed $RI$ during simulations of the corresponding optical properties at the interested incident wavelength. The real part ($Re$) of dust $RI$ is normally set to approximately 1.5, and the imaginary part ($Im$) is normally set between 0.001 and 0.01 at visible wavelengths, except for hematite and magnetite (Sokolik and Toon, 1999; Volten et al., 2001; Kandler et al., 2007; Meng et al., 2010). Previous studies using both measurements and numerical investigations reveal that a relatively large variation in the $RI$ of dust

materials in different regions or with different components does exist (Meng et al., 2010; Bi et al., 2011; Stegmann and Yang, 2017). Kemppinen et al. (2015) revealed that the retrieved dust $RIs$ based on the comparisons of scattering matrices between simulations and laboratory measurements deviate from the true dust $RIs$.

Figure 1 shows some examples of the $RIs$ at the visual wavelengths, and we illustrate values from various different studies, i.e., those from the GADS and the OPAC (referred to as well-accepted database values), the $RI$ spectrum based on

regions and dust components (Stegmann and Yang, 2017), the estimated values given by the Amsterdam-Granada Light

Scattering Database (AGLSD, Muñoz et al., 2000; Muñoz et al., 2012; Volten et al., 2001; Volten et al., 2006), and the assumed values from different modeling studies (Bi et al., 2010; Ishimoto et al., 2010; Merikallio et al., 2011; Liu et al., 2013; Kemppinen et al., 2015). The *Re* values are mainly between 1.4 and 1.7, and the range of *Im* values cover a few orders from 0.00001i to 0.1i. The uncertainty of the *RI* is obvious, and a better understanding of the *RI* is an important prerequisite

for studying the optical properties of dust and its further application.

This study uses the measured dust scattering properties from the AGLSD as the reference to 'evaluate' the modeled results. However, we will not pay too much attention to the effect or performance of different geometric factors, as this topic has been covered in several previous studies. Considering the obvious uncertainties and less attention related to the particle *RI*, this study introduces the important roles of the dust *RI* in developing corresponding models for the numerical simulation

of their optical properties. Section 2 introduces the models considered and the computational methods applied in this study. Section 3 investigates the impacts of the *RI* on the reproduction of the optical properties of several dust particles. The effects of the *RI* on model development are revealed in Section 4, and the impacts of the *RI* and geometries on dust optical properties are compared. Section 5 concludes the work.

## 2 Methodology

This study focuses on the effects of the *RI* on modeling the dust scattering matrix elements. The Koch-fractal particle has been used as the presumed geometry (Macke et al., 1996; Falconer, 2004; Liu et al., 2013; Jin et al., 2016). The Koch-fractal particle geometry has been used to produce concave polyhedra based on tetrahedron elements of different generations and is flexible in representing both the particle overall geometries and their detailed surface structures. Macke et al. (1996) used second-generation Koch-fractal particles with different irregularities to explore the scattering properties of complex ice

crystals using a geometric-optics method. Liu et al. (2013) extended the applications of Koch-fractal particle geometries to mineral dust particles and found that the corresponding optical properties represent those from measurements.

Figure 2 shows three examples of third-generation Koch-fractal particle geometries. The sequential number of the Koch-fractal generation indicates the complexity of the surface structure, and an irregular ratio defines the random



movements of higher-order polyhedra to specify particle irregularity. Liu et al. (2013) introduced the aspect ratio (*AR*, the ratio of height to width) to generate prolate or oblate particles. Macke et al. (1996) and Liu et al. (2013) include more details on the definition of the Koch-fractal particles. In Figure 2, the Koch-fractal particle on the left is the nearly regular particle with an *AR* of 1.0 (1.06 for a regular particle) and an irregular ratio of 0. The middle particle has an irregular ratio of 0 but an

*AR* of 2.5, and the particle on the right, which is the most irregular particle, has an *AR* of 2.5 and an irregular ratio of 0.3. Previous studies indicate that geometries with proper irregularity have better potential to characterize the geometric features of several types of actual dust particles. To constrain the geometric variations considered, all particles will be the third-generation Koch-fractal particles with an irregular ratio of 0.3, and we will only consider different *AR*s in this study.

To account for the irregular geometries, multiple numerical models are available to calculate the single-scattering

properties of nonspherical particles (Yang and Liou, 1996a; Mishchenko et al., 1997; Yurkin and Hoekstra, 2011; Bi et al., 2013). Following Liu et al. (2013), this study uses a combination of the pseudo-spectral time domain method (Liu, 1997; Liu et al., 2012) and the improved geometric-optics method (Yang and Liou, 1996b; Yang and Liou, 1998; Bi et al., 2014) to cover the required range of dust size parameters at visible incident wavelengths. For the irregular fractal particles, we use *r*, i.e., the radius of a volume-equivalent sphere, to define their sizes, so the size parameter *x* is defined as $x=2\pi r/\lambda$ (with $\lambda$ being

the wavelength). Computations are carried out to cover the size distributions of several typical dust particles. After integration of the optical properties over the simultaneously measured dust size distributions (Volten et al., 2006; Muñoz et al., 2012), the resulting bulk scattering matrix elements of certain *RIs* and particle geometries can be compared with the AGLSD measurements, and the agreements between the measurements and simulations are used to specify the potentials of the corresponding methods.

The square values of the differences between the simulated and measured results are used as the indicator to quantify the differences between the simulations and observations. For $P_{11}$, the difference $d_{11}$ is defined as Eq. (1):

$$d_{11} = \sum_{\theta=5°}^{173°} \left( \frac{P_{11}^{mea}(\theta) - P_{11}^{simu}(\theta)}{P_{11}^{mea}(\theta)} \right)^2, \qquad (1)$$

where $P_{11}^{mea}(\theta)$ is the measured $P_{11}$ element at scattering angle $\theta$ and $P_{11}^{simu}(\theta)$ is for simulated model. Note that the AGLSD provides the scattering matrix elements with scattering angles between 5° and 173°. Moreover, the difference $d_{ij}$ of


the other five scattering phase matrix elements can be given by Eq. (2):

$$d_{ij} = \sum_{\theta=5°}^{173°} \left( \frac{P_{ij}^{mea}(\theta) - P_{ij}^{simu}(\theta)}{P_{11}^{mea}(\theta)} \right)^2,$$
(2)

where $P_{ij}^{mea}(\theta)$ and $P_{ij}^{simu}(\theta)$ are the measured and simulated elements, respectively. The numerical model that gives the smallest $d_{11}$ will be defined as our optimal model for each dust sample, which also results in the minima of $d_{12}$, $d_{43}$, and $d_{44}$

for most cases, and $d_{22}$ and $d_{33}$ are very close to their minima. We have also tried to find the minimum of the $d_{ij}$ summation, i.e., $d_{11} + d_{12} + d_{22} + d_{33} + d_{43} + d_{44}$, as the optimal case, and very close results (a difference on $Re$ of 0.1 or a difference on $Im$ of $10^{-0.5}$) were obtained.

The scattering phase matrices of feldspar, quartz, loess, Lokon (volcanic ash), and red clay from the AGLSD are considered as the references, and again, this study emphasizes the role of the $RI$ of dust. These actual dust particles have

different compositions, shapes, and size distributions, which cause their unique optical properties. Therefore, by considering multiple kinds of dust particles, we attempt to demonstrate that the effects of the $RI$ generally hold. Furthermore, the influences of particle geometries can hardly be isolated or avoided during the study, so the roles of the $RI$ and geometry will be compared.

## 3 The impact of the refractive index

As discussed in Section 1, large variations in the $RI$s at visible wavelengths do exist for dust particles in different regions due to the differences in their components. To take advantage of the numerical investigation, we consider relatively larger ranges of $RI$s. The $Re$ ranges from 1.4 to 2.2 in steps of 0.1 (9 values), and values from $10^{-4}$ to $10^{-2}$ in steps of $10^{-0.5}$ (in logarithmical scale, i.e., 5 values) are used for the $Im$. We mostly focus on the optical properties at an incident wavelength of 633 nm, and the spectral consistency will be briefly discussed at the end of Section 3.

Figure 3 illustrates the bulk scattering matrices of simulated Koch-fractal particles with different $Re$ values. Third-generation Koch-fractal particles with an $AR$ of 2.5 and an irregular ratio of 0.3 are applied. Figure 3 clearly shows that different scattering matrix elements have distinct sensitivities to the changes in the $Re$. The effects of $Re$ on the $P_{11}$ /$P_{11}(30°)$ element mainly appear in the sideward and backward directions, and the scattering at scattering angles larger than

30° becomes stronger as $Re$ increases. With an increase in $Re$, the $P_{12}/P_{11}$ element becomes closer to zero, and the $P_{22}/P_{11}$

element departs from 1. The differences for the other three elements, i.e., $P_{33}/P_{11}$, $P_{43}/P_{11}$, and $P_{44}/P_{11}$, are less significant.

Another noteworthy phenomenon is that the differences between the computed scattering matrix elements with $Re$ between

1.4 and 1.6 are more obvious than those with $Re$ values of 2.0 and 2.2, illustrating that the scattering matrix elements are

more sensitive to the changes in $Re$ when the value of $Re$ is relatively small (e.g., 1.4).

Figure 4 is similar to Figure 3 but for results with different $Im$ values. $Im$ directly affects particle absorption, but its

impact on particle scattering properties cannot be ignored. We consider a dust sample with relatively larger sizes to better

demonstrate the effect of the $Im$. The bulk scattering matrix elements are obtained based on the size distribution of Lokon

samples, which have an effective radius of 7.1 μm. Again, different scattering matrix elements have distinct sensitivities to

the change in the $Im$, and the $P_{11}$, $P_{12}$, and $P_{22}$ elements are more sensitive to the $Im$. With increasing $Im$, the $P_{11}/P_{11}(30°)$

element decreases at scattering angles from 30° to 180°, indicating weaker side and backward scattering. The $P_{22}/P_{11}$ element

increases as the $Im$ increases. The $P_{33}/P_{11}$, $P_{43}/P_{11}$ and $P_{44}/P_{11}$ elements show less variation for different $Im$ values.

Additionally, the scattering matrix elements change in the same directions as the $Re$ increases or the $Im$ decreases.

Figures 3 and 4 clearly show the impacts of the $RI$s on the modeling particle scattering matrix elements. With the size

distribution measured simultaneously, the $RI$ and geometry both remain variables for the numerical studies. In contrast to

previous studies with a fixed $RI$ but with variable particle shapes (Merikallio et al., 2011, 2013; Ishimoto et al., 2010; Tang

and Lin, 2013; Bi and Yang, 2014; Nousiainen, 2014), this study tests whether the scattering matrix elements of different

dust aerosols can be reproduced by models with a fixed particle shape but different $RI$s.

Figure 5 compares the simulated scattering matrices of five dust species with measurements: feldspar, quartz, loess, red

clay, and Lokon (from left to right respectively). For the modeling results, a fixed geometry, i.e., the third-generation

Koch-fractal particle with an $AR$ of 2.5 and an irregular ratio of 0.3, is used for all simulations. Most of the previous

numerical and observational studies suggest that the $Re$ values lie between 1.5 and 1.6, and the scattering matrix elements are

more sensitive to the $Re$ change when the $Re$ values are relatively small. Therefore, we include an additional $Re$ of 1.55 in

this study, which results in a total of 50 complex $RI$s (10 $Re$ and 5 $Im$ values). The blue shaded regions in Figure 5 indicate

the variations in the simulated matrix elements with the 50 $RI$s, and the red curves correspond to the optimal cases that give



the minimum $d_{11}$ among the 50 cases. Even with a fixed geometry, the simulated results of the five dust samples can achieve reasonable agreement with the measurements, especially for the $P_{11}/P_{11}(30°)$, $P_{12}/P_{11}$, and $P_{33}/P_{11}$ elements. The modeling for feldspar, quartz, and loess are successful, and the optimal results for red clay slightly differ from the observations but still reflect the characteristics of red clay's scattering matrix. The computed results for Lokon particles achieve a relatively

accurate agreement with the measurements with a $Re$ larger than expected values, i.e., 2.2. Most of the $RI$s obtained for the optimal cases have a real part of 1.5-1.6 and an imaginary part between $10^{-4}$ and $10^{-3}$, consistent with generally accepted values and those suggested by the AGLSD. Table 1 lists the estimated $RI$s of five types of dust given by the AGLSD and the corresponding optimal $RI$s based on the particular geometry. It is also found that with an $RI$ that results in the minimum value of $d_{11}$, the minimum values of $d_{12}$, $d_{43}$, and $d_{44}$ are also obtained. Both the simulated and measured $P_{12}/P_{11}$ elements

show considerable variations, and the simulated results match the measurements by mainly changing the real part of the $RI$. However, the computed and measured $P_{22}/P_{11}$ elements show obvious differences. Both the simulated and measured $P_{33}/P_{11}$ elements show less variation, indicating that the $P_{33}/P_{11}$ element is less sensitive to the changes in the $RI$, and the simulated results with almost any $RI$ satisfactorily agree with the measurements. The $P_{44}/P_{11}$ elements show similar features to the $P_{33}/P_{11}$ elements, but the consistencies between the numerical results and the measurements are slightly worse than those of the

$P_{33}/P_{11}$ elements. Generally, the scattering matrices of different dust samples can be reproduced by applying proper $RI$s with a fixed geometry, although the differences between the simulated and measured quantity of some particular elements (e.g., the $P_{22}/P_{11}$ elements) are noticeable, which is the same as models considering geometric variations.

     Figure 6 compares the computed and measured scattering matrices of feldspar at two different incident wavelengths, in which the blue and red colors represent the results at incident wavelengths of 633 nm and 442 nm, respectively. Again, the

numerical results are based on the fixed Koch-fractal geometry and 50 different $RI$s, as mentioned above. The optimal numerical results show similar agreement with the measurements at the two wavelengths, as discussed above, i.e., close agreement for the $P_{11}/P_{11}(30°)$, $P_{12}/P_{11}$, $P_{33}/P_{11}$, and $P_{44}/P_{11}$ elements while relatively larger differences for the $P_{22}/P_{11}$ and $P_{43}/P_{11}$ elements. Furthermore, the spectral differences regarding the measured $P_{22}/P_{11}$ elements are not shown by the simulated results, while other elements show little spectral differences or agree with the simulations. The optimal cases for

both wavelengths correspond to the same $RI$ of $1.55+10^{-3}i$, which indicates relatively small wavelength independence of

feldspar *RI*. In other words, the Koch-fractal particle applied has a clear spectral consistency for modeling dust optical properties at multiple wavelengths, which can hardly be achieved by spheroid models (Merikallio et al., 2011; Dubovik et al., 2006; Lin et al., 2018). We also compare the spectral performance for the red clay and quartz scattering matrices, and similar results (the same optimal *RI* at the two incident wavelengths) are obtained.

However, the results for loess and Lokon are slightly different. Figure 7 illustrates the results for the loess sample as an example. The optimal results at wavelengths of 442 nm and 633 nm correspond to *RIs* of $1.8+10^{-4}$i and $1.6+10^{-4}$i, respectively. Additionally, the consistencies of the computed and measured results at the wavelength of 633 nm are slightly better than those at the wavelength of 442 nm, especially for the forward directions of the $P_{12}/P_{11}$ and $P_{43}/P_{11}$ elements. For Lokon, the same real part of *RI* (*Re* = 2.2) is obtained at the two wavelengths, while the imaginary parts are slightly different

($10^{-3}$i at 442 nm and $10^{-3.5}$i at 633 nm). This indicates that loess and Lokon may have stronger spectral differences with respect to their optical properties, which have to be considered for downstream radiative studies and remote sensing applications.

## 4 Refractive index *vs.* geometry

The importance of particle geometry in modeling dust optical properties has been well studied (Bi et al., 2010; Osborne et al.,

2011; Lin et al., 2018), and Section 3 indicates the clear role of the *RI*. Thus, with particle size relatively well constrained, it becomes interesting to investigate whether the geometry or *RI* plays the same or different roles in modeling the dust optical properties for remote sensing and radiative forcing studies.

     We first test whether different presumed *RIs* influence the determination of particle geometries for developing dust optical models. Figure 8 gives the measured and simulated scattering matrix elements of quartz, and the simulated results of

Koch-fractal particles with three different geometries (different *ARs* only) and two different *RIs* are illustrated. The particles with *ARs* of 0.25, 1.0, and 3.0 and the *RIs* of $1.5+10^{-3}$i (the red curves) and $1.7+10^{-3}$i (the blue curves) are used. If the quartz *RI* is assumed to be $1.5+10^{-3}$i for the numerical simulations, the modeled results based on the Koch-fractal particles with an *AR* of 1.0 agree most closely with the measurements (for almost all six elements), and those with larger (3.0) or smaller (0.25) *ARs* both depart from the measurements. However, if the *RI* is assumed to be $1.7+10^{-3}$i, then the results based on the particles



with *AR*s of 0.25 and 3.0 agree more closely with the measurements, except for the $P_{22}/P_{11}$ element. Clearly, Figure 8 indicates that if different *RI*s are assumed, then different geometries must be applied to represent the scattering properties of actual aerosols. Similar results are obtained for the loess and red clay samples as well (not shown here). These comparisons illustrate that the *RI* can significantly influence the determination of appropriate geometries in modeling studies of dust optical properties.

Furthermore, we directly compare the roles of the *RI* and geometry in reproducing dust scattering matrices. To assess the effects of the geometry, third-generation Koch-fractal geometries with different *AR*s and a fixed irregular ratio of 0.3 are tested. Because the *AR* shows the most significant influences on the scattering properties for our Koch-fractal particles, 10 different *AR*s (0.25, 0.5, 0.75, 1.0, 1.5, 2.0, 2.5, 3.0, 3.5, and 4.0) are considered in the tests. The *RI* is fixed at $1.6 \pm 10^{-3.5}$i (close to the *RI*s of feldspar, quartz, loess, and red clay obtained for Figure 5). For comparison, the effects of different *RI*s will be illustrated by considering particles with the fixed geometry used above.

Figure 9 illustrates the scattering matrices of Lokon as well as the simulated results for particles with different *AR*s and *RI*s. The blue curves indicate the optimal case (*AR*=1.0) among those with different *AR*s and the same *RI*, and the red curves are for the optimal case with an *RI* of $2.2 + 10^{-3.5}$i among those with different *RI*s. First, we discuss the results for $P_{11}$. The results for particles with different geometries, i.e., *AR*s, but a fixed *RI* are illustrated by the blue areas. With the *RI* close to the widely accepted values, the results based on any geometry differ from the measurements, especially for the values with scattering angles larger than 90°. However, the red areas can cover the measurements at the edge and indicate that simulated results with an extreme *RI* can better reproduce the scattering matrix of Lokon. For other elements, the optimal results, i.e., the red and blue curves, give similar agreement with the measurements, especially for the $P_{33}/P_{11}$, $P_{43}/P_{11}$, and $P_{44}/P_{11}$ elements. Overall, for the Lokon measurements, a more reasonable *RI* has to be used to reproduce their optical properties, and this may be the reason why few results on modeled Lokon samples have been previously published. The other features illustrated by Figure 9 are the coverage differences among the red and blue regions, which indicate the sensitivities of the particle geometry and *RI*. For the $P_{11}$, $P_{12}/P_{11}$, and $P_{22}/P_{11}$ elements, the red areas clearly cover the blue ones, indicating that particles with different *RI*s result in larger variations in the corresponding elements than those with different geometries. The $P_{33}/P_{11}$ and $P_{43}/P_{11}$ elements for particles with either different geometries or different *RI*s show similar coverages, while the

particle geometry may lead to larger variations in the $P_{44}/P_{11}$ elements. These differences may become useful during the development of numerical dust optical models based on observed results such as the AGLSD.

Figure 10 is similar to Figure 9, except for the red clay particles. The blue curves correspond to results with an *AR* of 0.25, and the red curves correspond to those with an *RI* of $1.8+10^{-2}i$. Comparing the two optimal cases within the different *AR*s and *RI*s, the optimal results among particles with different *AR*s achieve a better consistency with the measurement for $P_{11}$, $P_{22}/P_{11}$, $P_{33}/P_{11}$, and $P_{44}/P_{11}$, whereas the $P_{12}/P_{11}$ results from the *RI* optimal case is slightly better. The $P_{43}/P_{11}$ element from both cases differs from the measurements. For this case, the agreement between the modeled and measured results is clearly improved by changing the particle geometry. Similar to Figure 9, Figure 10 also illustrates that the changes in the geometry and *RI* can provide the simulations for different scattering matrix elements with different degrees of variation.

For the results of the feldspar particles in Figure 11, the most notable feature is that the two optimal cases with different variables are highly consistent; the reproductions of the matrix elements (except $P_{22}/P_{11}$) are quite successful, and the simulated $P_{22}/P_{11}$ elements of the two optimal cases both deviate from the measurement with the scattering angles between 60° and 140°. For feldspar, i.e., the most extensively studied dust sample among the AGLSD (Bi et al., 2009; Dubovik et al., 2006; Liu et al., 2013; Lin et al., 2018; Merikallio et al., 2011; Volten et al., 2001), multiple models with appropriate combinations of the particle *RI* and geometry all result in close agreement with measurements. Reasonable results can be obtained by merely changing the *RI* even if the geometry is relatively different from reality and vice versa.

Although the geometry and *RI* cannot be directly compared, Figures 9-11 still briefly demonstrate their differences in influencing dust scattering matrix elements, and we find that the *RI* plays a role as important as that of the geometry for the development of dust optical models. However, considering the irregular dust geometries, there may be countless geometry models, whereas the *RI* appears to be a more manageable parameter. Additionally, a proper *RI* is of great importance to the optical propertie simulations, while the development of geometries can be time-consuming but ineffective for some kinds of dust particles.

## 5 Conclusions

This study investigates the role of the *RI* in modeling the dust scattering matrix elements. Instead of reproducing the dust

scattering properties by building one or a group of nonspherical geometries at a fixed *RI* (that may or may not be an accurate *RI* due to its uncertainties), we emphasize the sensitivities of the scattering phase matrix on the particle *RI*. By simply changing the *RI* during the numerical modeling, it is possible to characterize the optical properties of different dust particles, even if the model geometry is fixed. As a result, it becomes possible to simplify the model developments for different

mineral dust particles. To be more specific, instead of constructing and testing various geometric models, using results from particles with different *RI*s but a fixed geometry can also be a solution to calculate scattering properties of dust particles at different wavelengths, which would be more flexible and computationally efficient.

As expected, if different *RI*s are considered for dust optical property simulations, the appropriate geometry that leads to the best agreement to the observation will change accordingly. By comparing the sensitivities of the dust scattering matrix

elements, it is noticed that the reproductions of the scattering matrix elements of different dust particles respond differently to the change in the *RI* and geometry. With a known particle size distribution, the scattering matrix of some kinds of dust (e.g., feldspar, quartz, and loess) can be well reproduced by adjusting either the *RI* or the geometry with the other parameters fixed, but those of other dust particles (e.g., red clay and Lokon) can only be reproduced by applying an extreme and fixed geometry or *RI*. As a result, more efforts should also be devoted to better constraining the particle *RI* during the development

of aerosol optical properties for remote sensing and radiative transfer applications.

*Author contributions.* YH and CL designed the study, carried out the research, and performed the numerical simulation. YH, CL, YY, and LB discussed the results and wrote the paper. All authors gave approval for the final version of the paper.

*Competing interests.* The authors declare that they have no conflict of interest.

*Acknowledgements.* We thank the Amsterdam-Granada Light Scattering Database for providing the measured data on the geometric and scattering properties of dust. This work was financially supported by the National Natural Science Foundation of China (NSFC No. 41571348) and the Natural Science Foundation of Jiangsu Province (BK20190***).



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

Table 1: The estimated refractive index (*RI*) values given by the Amsterdam-Granada Light Scattering Database (AGLSD) and the *RI*s corresponding to our optimal numerical results.

|  | Feldspar | Quartz | Loess | Red clay | Lokon |
|---|---|---|---|---|---|
| *Re* in the AGLSD | 1.5-1.6 | 1.54 | 1.5-1.7 | 1.5-1.7 | 1.5-1.6 |
| *Im* in the AGLSD | $10^{-3}$-$10^{-5}$ | 0 | $10^{-3}$-$10^{-5}$ | $10^{-3}$-$10^{-5}$ | $10^{-3}$-$10^{-5}$ |
| Optimal *RI*[a] | $1.55+10^{-3}i$ | $1.6+10^{-4}i$ | $1.6+10^{-4}i$ | $1.8+10^{-2}i$ | $2.2+10^{-3.5}i$ |

a.   The optimal *RI* values were based on the Koch-fractal particle geometry and the 50 different *RI* values considered in this study.



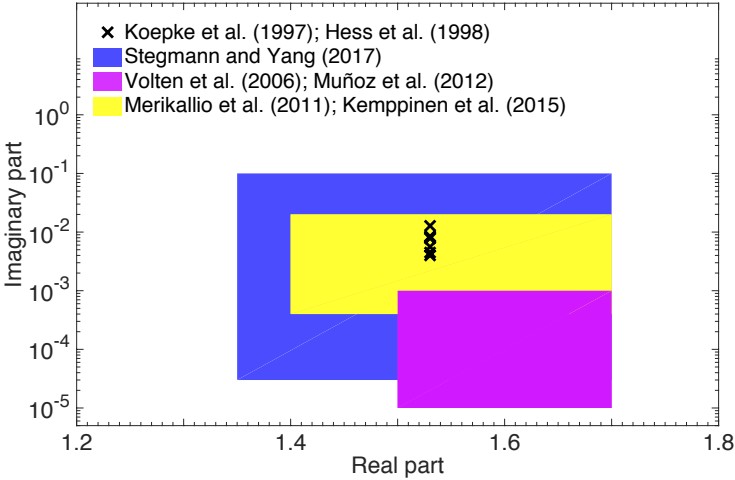

**Figure 1: Refractive index from different sources.**



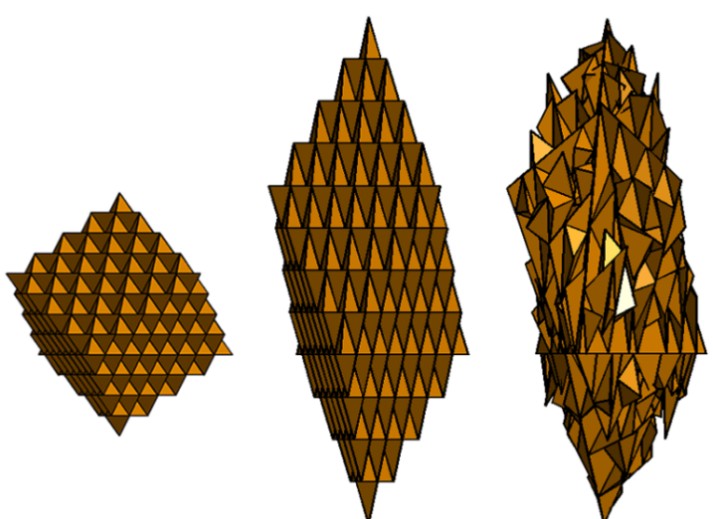

**Figure 2: Third-generation Koch-fractal particles with different geometric parameters. The aspect ratios of the particles from left to right are 1.0, 2.5, and 2.5, and the irregular ratios are 0, 0, and 0.3, respectively.**




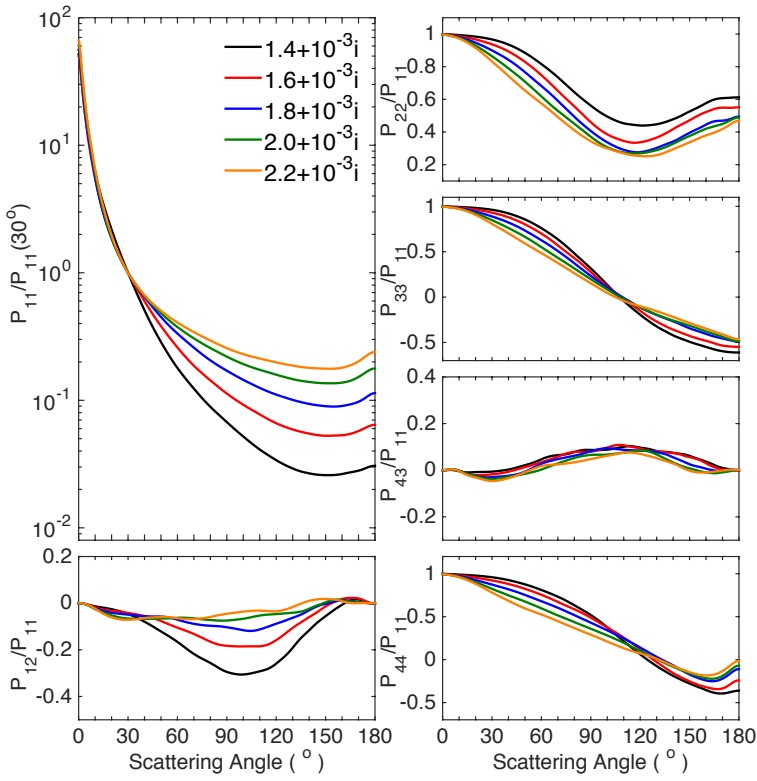

**Figure 3: Simulated scattering matrix elements for Koch-fractal particles with different refractive indices. Here, the imaginary part is fixed at 0.001i, and the real part is changed from 1.4 to 2.2 with a step of 0.2. The particles are assumed to have the size distribution of feldspar, and the optical properties are simulated by considering an incident wavelength of 633 nm.**





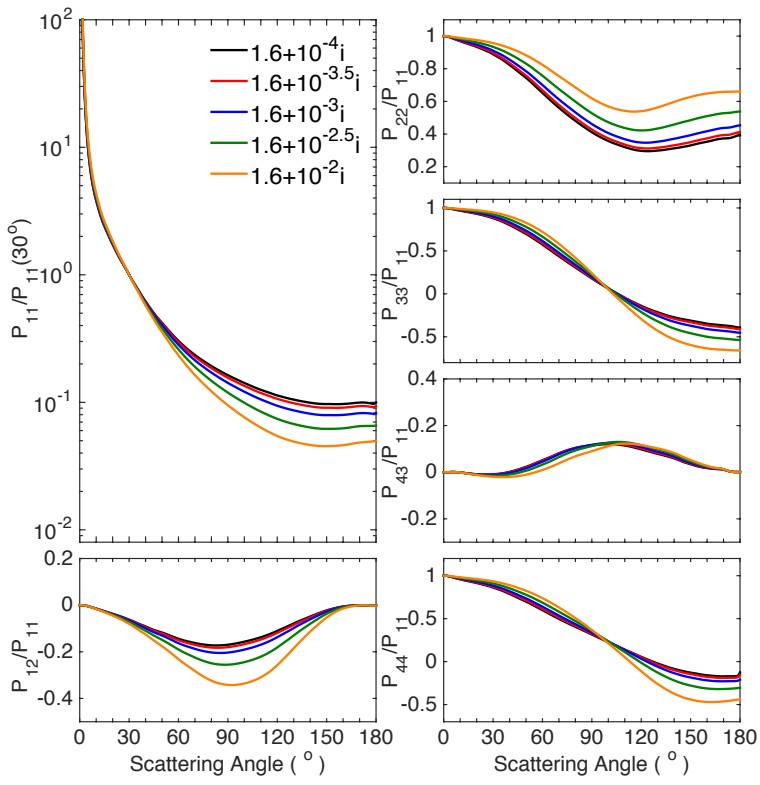

**Figure 4: Similar to Figure 3, except that the real part is fixed at 1.6, the imaginary part is changed from $10^{-4}$ to $10^{-2}$ with a step of $10^{0.5}$, and the particles are assumed to have the size distribution of Lokon.**






**Figure 5: Comparison of the scattering phase matrix elements of feldspar, quartz, loess, red clay and Lokon particles with different refractive indices between laboratory measurements and computed results at a wavelength of 633 nm. The optimal cases are also shown as the red curves.**





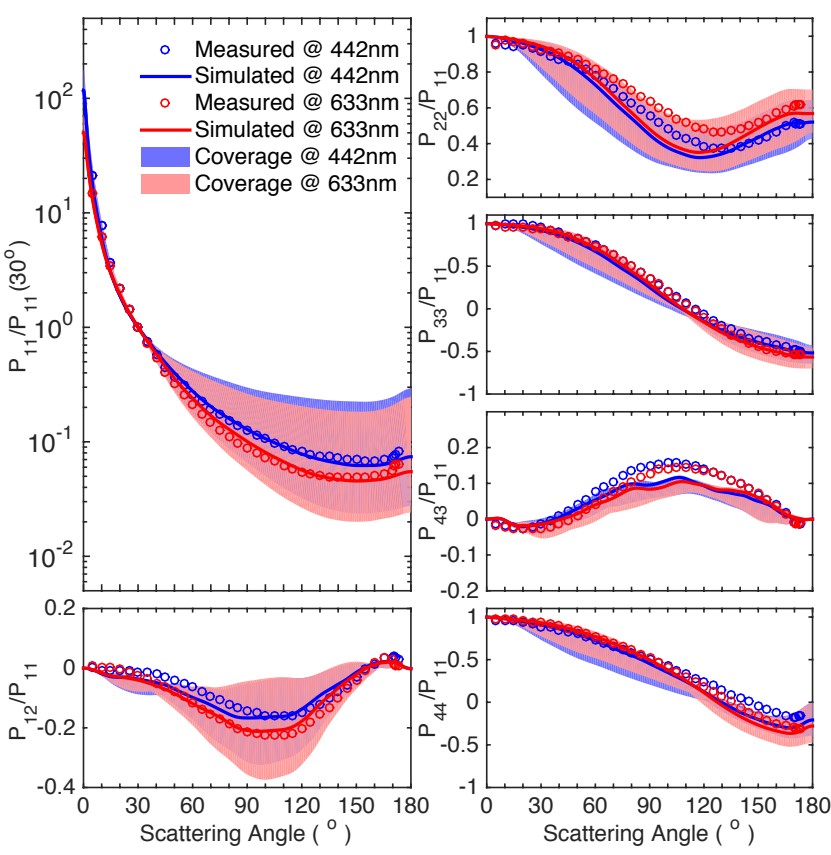

**Figure 6: Comparisons of the scattering matrix elements between the measured and computed results for feldspar at wavelengths of 442 nm and 633 nm. The hallow dots indicate the measurement, the thick lines indicate the optimal computed results, and the shaded areas indicate the variety range by different *RI*s. The optimal *RI* at both wavelengths is 1.55+10$^{-3}$i.**





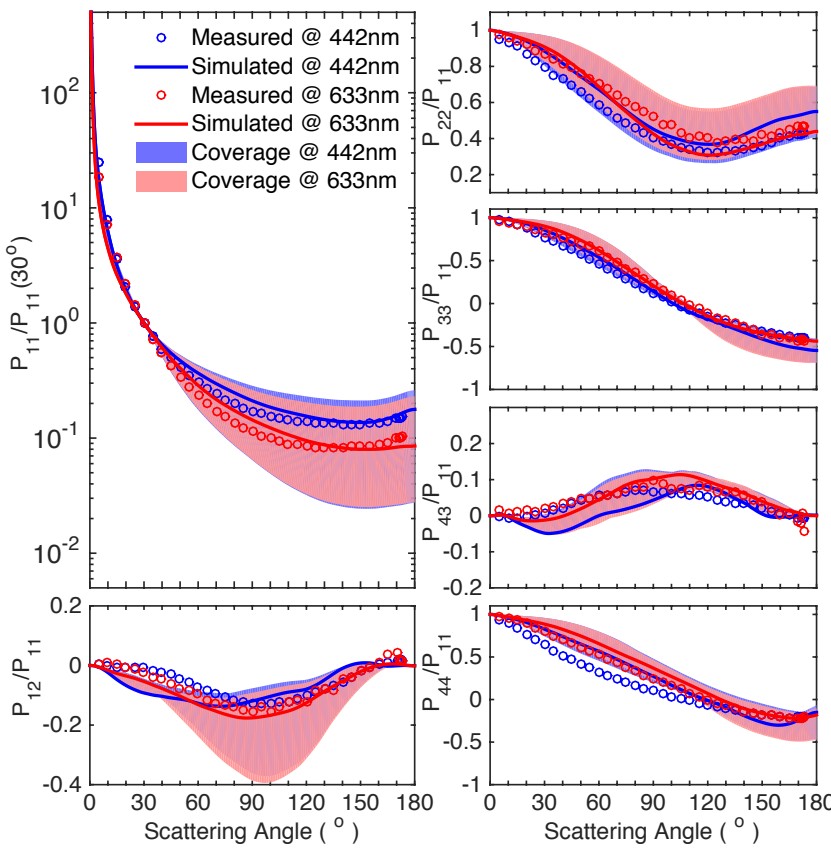

**Figure 7: Similar to Figure 6, but for the loess sample. The optimal *RI* is 2.2+10⁻²i at the wavelength of 442 nm and 1.6+10⁻⁴i at the wavelength of 633 nm.**




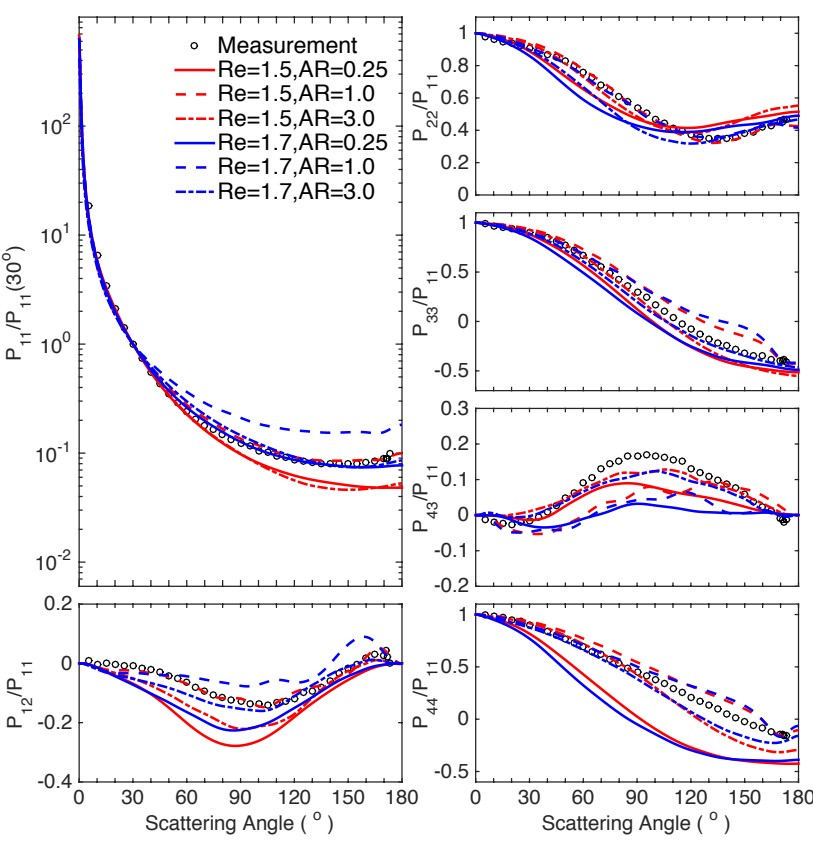

**Figure 8: Comparison of the scattering matrix elements between the simulations and measurements for quartz samples with three different particle ARs at two different *RI*s: 1.5+10⁻³i and 1.7+10⁻³i. The incident wavelength is 633 nm.**

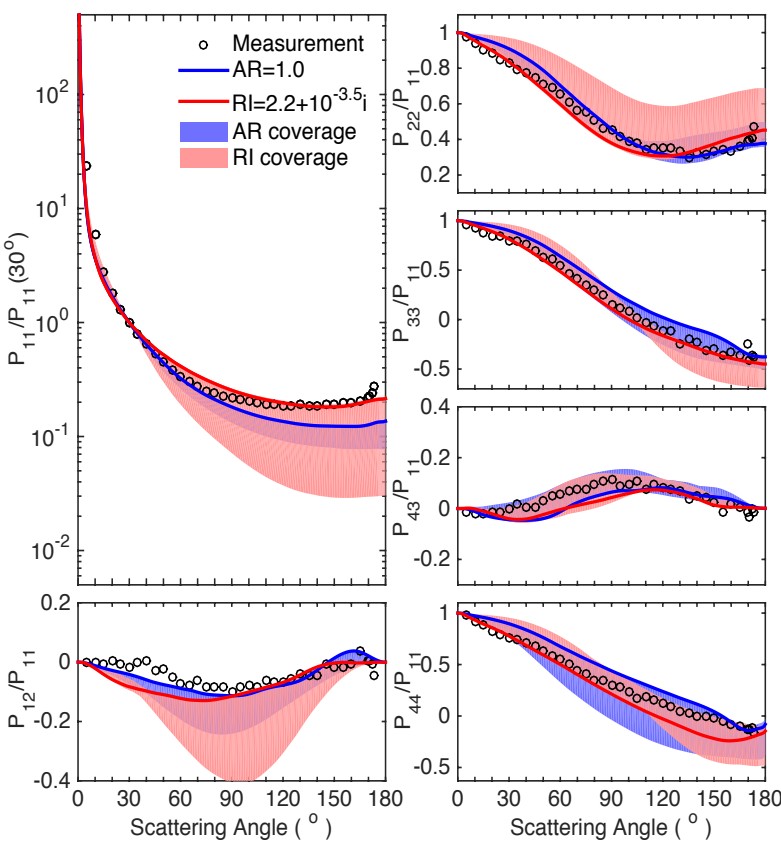

**Figure 9: Comparison between the simulated results of the scattering phase matrix elements of the Lokon samples with different variable parameters, including coverage with the AR to be the variable (shaded blue area), coverage with the *RI* to be the variable (shaded red area), optimal case for the two variable parameters (blue and red lines) and the measurement given by the AGLSD (black hallow dots).**





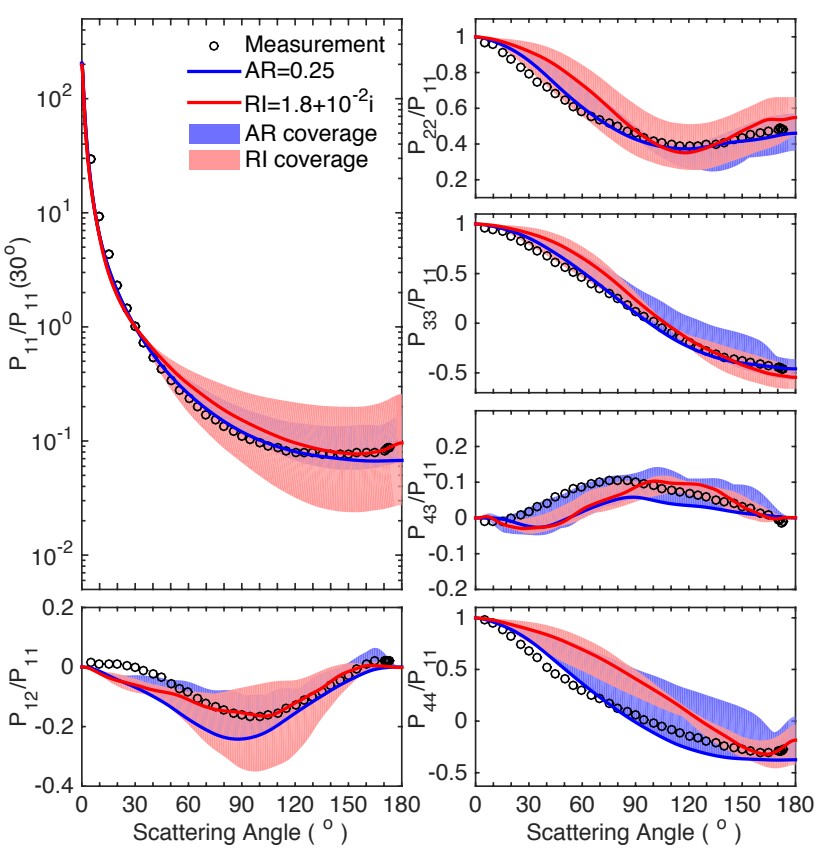

Figure 10: Similar to Figure 9 but for the red clay sample.





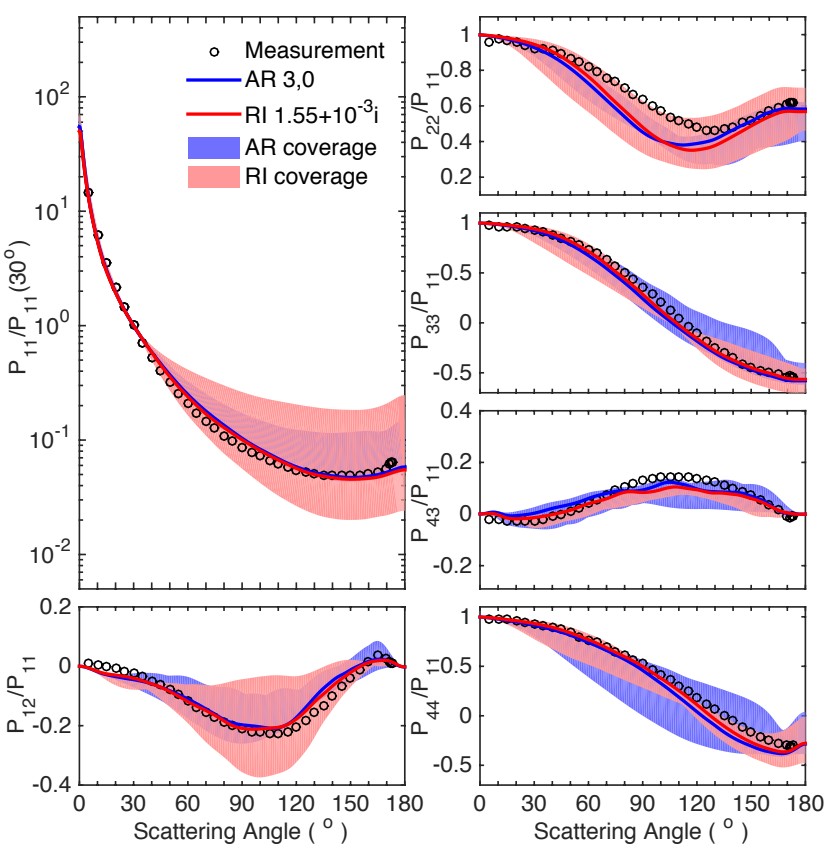

**Figure 11: Similar to Figure 9 and 10 but for the feldspar sample.**