# Peer review of "Scattering matrices of mineral dust aerosols: a refinement of the refractive index impact"

_Atmospheric Chemistry and Physics, 2019_

## Referee Comment (RC1) · Anonymous Referee #1 · 11 Nov 2019

Aerosol optical properties are fundamental for our understanding of their radiative effects in the realm of remote sensing. This manuscript investigated the optical properties of dust aerosols, with a focus on the role of the refractive index and its influence on current model development. By evaluating models with laboratory study of scattering matrix, the authors found the refractive index is as important as the particle shape in determining dust models. The current results shown in this paper support the conclusion that refractive index should be considered more carefully in studies of aerosol radiative effects. Therefore, the subject and the contents of this paper are interesting, and it might contribute to our further understanding of the behavior of various aerosols not just limited to dust. The paper in current status is well organized and nicely written, but I still want to raise a few questions before it gets published:
1. Line 23, Page 3: Figure 1 shows the large variations on dust refractive indices. However, it is unclear why do the authors need to use both crosses and shaded areas for their illustration? Can the authors be more specific on their motivations based on those studies?

2. Line 9, Page 5: It seems that a couple of important parameters for these numerical simulations are not introduced in the method section. For example, what is the range of sizes on dust particles considered for the numerical simulations? How to take into account the particle orientation during simulations?

3. Line 20, Page 5: In Section 2, the authors use the summation of relative errors of the six non-zero scattering matrix elements to specify the "accuracy" of the numerical model. However, bear in mind that different elements might have different variations. Thus, the relative errors may have quite different magnitudes, which could make the evaluations might not be that fair. Meanwhile, some mentioned studies only considered the relative errors of the scattering phase function, which also makes the comparison not purely apple-to-apple. I am wondering how different the results will be if different variables were considered?

4. Line 19, Page 7: Figure 5 illustrates very informative scattering phase matrices for the five dust samples. As we all can see, the numerical results achieve quite a different accuracy and different refractive indices. I am not sure I am entirely clear of the causes of the differences, and I hope the author could provide more thorough discussions in revision.

5. Line 19, Page 7: Comparing to Figure 5, the P22 appears to be the worst (among all six elements) comparing the model simulation and the observations. Why is that? What can further be done to limit this discrepancy here?

6. Line 6, Page 9: The authors mentioned that Figure 7 is the optimal simulation results with RI of $1.8+10^{(-4)}i$, but the caption mentioned $1.6+10^{(-4)}i$.

7. Lines 17-22, Page 11: The last paragraph of Section 4 is quite confusing for me. Actually, the comparisons in Figures 9-11 as well as the corresponding discussions before this paragraph are quite clear.

8. The authors mainly considered the differences in particle shapes and RI. Inevitably, the aerosol particle size could be another key variable here. Did the author do any simulation on the effect of sizes? This could complicate the comparisons tremendously, but it is worth to show only the most apparent changes when size is taken into account.

9. "Scattering matrix" and "phase matrix" are both used in the manuscript, but indeed they represent different physical quantities.

10. Line 24, Page 2: It should be "spheroids, ellipsoids, and superellipsoids" instead of "a spheroid, an ellipsoid, and a superellipsoid"

11. Lien 24, Page 3: "referred to as well-accepted database values" is inconsistent with the label in the figure.

12. Line 5, Page 4: If I understand it correctly, the aspect ratio refers to the proportional relationship between particle height and its width. So a larger aspect ratio means the particle is larger in height but relatively smaller in width. Then, how to comprehend the irregular ratio of 0.3, for example? How is irregular ratio defined?

---

## Short Comment (SC1) · 19 Dec 2019

**Comments for:**

"Scattering matrices of mineral dust aerosols: a refinement of the refractive index impact" by Yifan Huang, Chao Liu, Yan Yin, Lei Bi

**General comments:**

This manuscript provides a good overview on the current state of mineral dust single scattering computations and addresses an important issue in the field of aerosol optics, namely the uncertainty in the particle refractive index.

Is there any specific justification for the Koch-fractal morphology compared to e.g.

spheroids and super-ellipsoids? Is it more convenient or more accurate?

It would be interesting (but by no means necessary) to supplement this study with the derivatives of the phase matrices w.r.t. the refractive index.

**Specific Notes:**

- page 1, line 10: Change "Dust" to "Mineral dust".

- page 1, line 14: "This study reveals the importance of the dust RI for the model development of dust optical properties"

- Introduction in general: The IPCC has also identified aerosols as a major source of uncertainty in radiative forcing of the terrestrial climate. Maybe this can be added to the introduction with a suitable source to cite.

- p.2, l.4: "... mineral dust is widely distributed around the globe, ..."

- p.2, l.13: "For example, the measured phase functions of dust particles are distinctively different from the ...."

- p.2, l.20: " a simplified but optically equivalent model is more convenient and easier to process"

- p.3, l.14f: "..., to which much less attention has been devoted during the model development."

- p.5, l.15ff: Particle size distributions are a source of uncertainty as well. What is the reason for chosing the specific particle size distributions here?

- p.6: The proposed method finds the optimal theoretical particle properties in terms of the phase matrix alone. What is the impact on the other properties, such as extinction coefficient, albedo etc.? Is a similarly good match achieved?

---

## Referee Comment (RC2) · Anonymous Referee #2 · 14 Jan 2020

This paper by Huang et al. reveals by model simulations the importance of the dust refractive indices (RIs) for the model development of its optical properties. They show that the scattering matrix elements of different kinds of dust particles can be reasonably reproduced by choosing appropriate RIs even using a fixed particle geometry and that a change in the RI can strongly affect the appropriate shape parameters to reproduce the measured dust phase matrix elements. The study indicates that the development of corresponding optical models can potentially be simplified by considering only variations over different RIs. The study should be a welcome addition to the literatures on modeling and measurements of dust optical properties and their radiative effects. The paper is well written in general, though the model used needs to be introduced more specifically and the parameters presented to be described more clearly. I would
recommend the paper be published after minor revisions.

Minor/Technical issues:

P2, L5: What implications of this study for better quantifying these two fundamental parameters? This issue may also need to be highlighted in the conclusions.

P2, L7: There are two literatures given in the References corresponding to Xu et al., 2017 here.

P2, L10: There are two literatures given in the References corresponding to Bi et al., 2018 here.

P3, L20: There are two literatures given in the References corresponding to Bi et al., 2011 here.

P4, I11: Several types of dust particles?

P6, Eq. (2): What are i and j stand for, respectively? Their ranges should be given in the equation. Should it be Pij in the denominator?

P6, L6: Is this summation really used in the following sections? If so, an equation might be given here.

P7, L1-2: Do you mean element(s) or element ratio(s) here?

P7, L17: the literature for Nousiainen, 2014 is missing in the References.

P8, L1-2: Where is d11 shown in the figure? The phrase "element ratios" might be more suitable for P11/P11(30 degree), P12/P11, and P33/P11? Are all P11 in the denominator for 30 degree? Since these ratios are frequently used, their definitions (or meanings) need to be given in Sect. 2.

P14, L2-3, L16-17, and L24-25; P16, L3-4; P17, L5-6 and L10-12; P18, L14-15 and L16;

P19, L1-2 and L5-7: Are these literatures referred in the main-body text?
P22, Fig. 3 and P23, Fig.4: Is P11 for 30 degree in all the Y-axis?

P24, Fig.5: Are measurements referred by black dots? Legends (or descriptions) of the plots need to be given.

---

## Author Comment (AC1) · 19 Jan 2020

**Responses to Anonymous Reviewer #1 (Manuscript # acp-2019-812)**

First of all, we would like to thank the anonymous reviewer for his/her thoughtful review and valuable comments to the manuscript. In the revision, we have accommodated all the suggested changes into consideration and revised the manuscript accordingly. All changes are highlighted in RED in the revision. In this point-to-point response, the reviewer's comments are copied as texts in BLACK, and our responses are followed in BLUE.

General comment:

Aerosol optical properties are fundamental for our understanding of their radiative effects in the realm of remote sensing. This manuscript investigated the optical properties of dust aerosols, with a focus on the role of the refractive index and its influence on current model development. By evaluating models with laboratory study of scattering matrix, the authors found the refractive index is as important as the particle shape in determining dust models. The current results shown in this paper support the conclusion that refractive index should be considered more carefully in studies of aerosol radiative effects. Therefore, the subject and the contents of this paper are interesting, and it might contribute to our further understanding of the behaviour of various aerosols not just limited to dust. The paper in current status is well organized and nicely written, but I still want to raise a few questions before it gets published.
**Response:** Thanks for the constructive comments. The comments on the methods as well as the result interpretations are significantly helpful to improve the manuscript, and make the paper more solid. The following presents our point-to-point responses as well as the revision for the manuscript.

**Major Comments:**
1. Line 23, Page 3: Figure 1 shows the large variations on dust refractive indices. However, it is unclear why do the authors need to use both crosses and shaded areas for their illustration? Can the authors be more specific on their motivations based on those studies?
**Response**: Figure 1 has been significantly improved to better illustrate the refractive

indices from different measurements and literatures (see the following updated Figure). In the new figure, the dots represent the values of dust refractive indices given by the Global Aerosol Data Set (GADS; Koepke et al., 1997) and Optical Properties of Aerosols and Clouds (OPAC; Hess et al., 1998), which share the same data source, and the dot colors from blue to red correspond to the wavelength from 200 nm to 1000 nm. The curves are refractive indices of particle mineral components from Stegmann and Yang (2017). The blue shaded area indicates the values of several types of dust refractive indices estimated by the Amsterdam-Granada Light Scattering Database (ALSD, Muñoz et al., 2012; Volten et al., 2001; Volten et al., 2006), and the red shaded area is an example for the refractive index range applied in a numerical study by Kemppinen et al. (2015).

Overall, the figure that considerable uncertainties do exist on the refractive indices of dust aerosols. However, most studies on aerosol optical properties treat the dust refractive indices as a fixed value, and it is really necessary and important to consider such variations on refractive indices for applications such as aerosol measurements, retrievals, or radiative forcing studies.

[Figure]

Figure 1. New version of the figure to illustrate dust refractive index variations.

2. Line 9, Page 5: It seems that a couple of important parameters for these numerical simulations are not introduced in the method section. For example, what is the range

of sizes on dust particles considered for the numerical simulations? How to take into account the particle orientation during simulations?

**Response**: Thank you for the important suggestions, we have clarified the parameters in the revision, and they include the following:

(1). Dust samples considered in this study include feldspar, quartz, loess, Lokon (volcanic ash), and red clay, and, as mentioned in the manuscript, particle sizes are measured simultaneously. The numerical simulation just covers the entire observed particle size range (i.e, from 0.076 μm to 105 μm). Because a combination of the PSTD and IGOM is used for optical properties simulations, we can cover the entire size range observed by the ALGSD.

(2) As mentioned in Section 2, two numerical methods are applied to compute the optical properties of the models. For the PSTD, 128 orientations (16 values for θ and 8 values for φ) are considered for each particle to give optical properties of randomly oriented particles, and 128 directions are enough to give relatively smooth results for the computed results. Liu et al. (2012) used 48 orientations on the computation with hexagonal column models, and Jin et al. (2016) used 128 orientations on the computation with Koch-fractal geometries. The Monte-Carlo-based IGOM directly gives optical properties of randomly oriented particles.

We have included these discussions in the revision as:

"*Simultaneous size measurements by the AGLSD have sample sizes ranging from 0.076 μm to 105 μm (Volten et al., 2006; Muñoz et al., 2012), so we perform numerical simulations within the same range. The pseudo-spectral time domain method is applied to deal with the optical properties of geometries with size parameters up to 30, and those with size parameters over 30 are calculated by the improved geometric-optics method (Liu et al., 2013). For the computations of the PSTD, the optical properties of randomly oriented particles are averaged over those from 128 different orientations, which result in relatively smooth scattering matrix elements.*"

3. Line 20, Page 5: In Section 2, the authors use the summation of relative errors of the six non-zero scattering matrix elements to specify the "accuracy" of the numerical model. However, bear in mind that different elements might have different variations. Thus, the relative errors may have quite different magnitudes, which could make the evaluations might not be that fair. Meanwhile, some mentioned studies only

considered the relative errors of the scattering phase function, which also makes the comparison not purely apple-to-apple. I am wondering how different the results will be if different variables were considered?

**Response**: Thanks for the valuable comment. It's true that the method applied is unable to find out one refractive index that makes the simulation achieves the best consistency for all six nonzero elements. We have clarified this as the following:

*"The numerical model that gives the smallest d will be defined as our optimal model for each dust sample. Actually, we also compared the differences among other scattering matrix elements, and the optimal case is mostly consistent with the one considering only P11. As a result, we try to keep the evaluation simple, and use only d as a criterion."*

4. Line 19, Page 7: Figure 5 illustrates very informative scattering phase matrices for the five dust samples. As we all can see, the numerical results achieve quite a different accuracy and different refractive indices. I am not sure I am entirely clear of the causes of the differences, and I hope the author could provide more thorough discussions in revision.

**Response**: We have improved the discussions related to Figure 5 as the following:

*"For feldspar sample, $P_{11}/P_{11}(30°)$, $P_{12}/P_{11}$, $P_{33}/P_{11}$, and $P_{44}/P_{11}$ of the optimal case agree closely with the measurements. Differences are only noticed for $P_{22}/P_{11}$ at the scattering angles from 60° to 150° and the $P_{43}/P_{11}$ from 75° to 150°. Similar results are obtained for quartz and loess samples. The optimal results for red clay sample are less consistent with the measurements when compared with the results for the three samples above. Certain deviations between the computed and measured results appear at the forward direction for every nonzero matrix element of red clay except for the $P_{11}/P_{11}(30°)$. Furthermore, RI of the corresponding optimal case for red clay sample is also obviously different from these discussed above, i.e. 1.8 for the Re and $10^{-2}$ for the Im. The computed results for Lokon particles achieve a relatively accurate agreement with the measurements with a Re much larger than expected values, i.e., 2.2. However, the reproductions of the forward directions of $P_{12}/P_{11}$ and $P_{43}/P_{11}$ are not satisfactory."*

5. Line 19, Page 7: Comparing to Figure 5, the P22 appears to be the worst (among all six elements) comparing the model simulation and the observations. Why is that?

What can further be done to limit this discrepancy here?

**Response**: We also noticed that the $P_{22}$ element show different characteristics compared with the other nonzero elements. For the $P_{22}$ element, a certain error exists between the computation and measurement even if the error has been minimized by applying a proper refractive index. Similar results are also shown by Tang and Lin (2013) and Lin et al. (2018) if results with different geometries are applied. Figure 10 showed that the $P_{22}$ element of red clay can be successfully reproduced by applying Koch-fractal particles with an aspect ratio of 0.25. Lin et al. (2018) also illustrated that $P_{22}$ element is sensitive to the aspect ratios while applying spheroid and super-spheroid geometries. These indicate that the $P_{22}$ element is strongly influenced by particle geometry, so there may be larger discrepancies. As a result, such discrepancy on $P_{22}$ may be limited by improving the particle geometry model in the future. We have clarified this in the revision.

6. Line 6, Page 9: The authors mentioned that Figure 7 is the optimal simulation results with RI of 1.8+10^(-4)i, but the caption mentioned 1.6+10^(-4)i.

**Response**: Sorry for the mistake. Results with a RI of $2.2+10^{-2}i$ at the wavelength of 442 nm give best agreement to the observations of loess samples. Both values in the manuscript are incorrect, and we have corrected them. Meanwhile, we have double checked all those values in the manuscript, and there should be no such typo.

7. Lines 17-22, Page 11: The last paragraph of Section 4 is quite confusing for me. Actually, the comparisons in Figures 9-11 as well as the corresponding discussions before this paragraph are quite clear.

**Response**: The last paragraph was originally presented to conclude the results shown by Figures 9-11. To be more specific, we try to emphasize the necessity of taking the uncertainty of RI into consideration in numerical studies of dust optical properties, but failed to present as clear as possible. Considering the suggestion of the reviewer, we have rewritten this paragraph as following:

   *"Obviously, both RI and geometry significantly affect mineral dust optical properties but quite differently, and, even without consideration of the influence of particle size, an accurate RI has to be determined to develop an appropriate dust geometric model, and vice versa. However, if only an optically equivalent model at a single wavelength or a limited number of wavelengths is required, our results indicate that either RI or*

*geometry can be treated as a variable while fixing the other. Thus, instead of constructing dust model by building different geometries (e.g., Mishchenko et al., 1997; Bi et al., 2010; Liu et al., 2012; Lin et al., 2018), it is also potentially possible to consider only results from a fixed particle geometry but with various RIs. The later (fixing a geometry and changing only RI) may be more convenient, because the RI can be defined more quantitatively.''*

8. The authors mainly considered the differences in particle shapes and RI. Inevitably, the aerosol particle size could be another key variable here. Did the author do any simulation on the effect of sizes? This could complicate the comparisons tremendously, but it is worth to show only the most apparent changes when size is taken into account.

**Response**: The optical properties of a particle is determined by its size, refractive index, and shape/geometry. The sensitivity of optical properties on size has widely been studied and well known, so we didn't intend to discuss the size effect in this study. Furthermore, the main purpose of this manuscript is to investigate the role of the RI in modeling the dust scattering matrix elements. The computed results are integrated according to the size distributions given by the measurements to ensure that the effect of size is eliminated. However, the impact of sizes is definitely an interesting topic for future studies.

9. "Scattering matrix" and "phase matrix" are both used in the manuscript, but indeed they represent different physical quantities.

**Response**: Thank you for the constructive comment. By definition, the phase matrix relates the Stokes parameters of the incident and scattered beams defined relative to their respective dimensional planes, and the scattering matrix relates the Stokes parameters of the incident and scattered beams defined with respect to the scattering plane, that is, the plane through the unit vectors (van de Hulst, 1957; Bohren and Huffman, 1983). Scattering matrices can reflect the different optical properties of various mineral dusts as all polarizing properties of the scatterers are contained in the scattering matrices (Volten et al., 2001). We discussed the scattering matrices in this manuscript, and have replaced all "phase matrix" by scattering matrix.

10. Line 24, Page 2: It should be "spheroids, ellipsoids, and superellipsoids" instead

of "a spheroid, an ellipsoid, and a superellipsoid"
**Response**: Thanks for the suggestion, and we have corrected the sentence.

11. Line 24, Page 3: "referred to as well-accepted database values" is inconsistent with the label in the figure.
**Response**: Corrected

12. Line 5, Page 4: If I understand it correctly, the aspect ratio refers to the proportional relationship between particle height and its width. So a larger aspect ratio means the particle is larger in height but relatively smaller in width. Then, how to comprehend the irregular ratio of 0.3, for example? How is irregular ratio defined?
**Response**: Thanks for the comment. Irregular ratio is a real number within the range [0, 0.5] to specify the random movement of the position of apex to generate irregular particles. The irregular ratio used to constrain the maximum movement the higher order apexes can move during the generation of the Koch-fractal particle. If it is 0, a regular particle is generated. If the irregular ratio become close to 0.5, the apexes can be randomly moved to a much wider range. See the following figure. We briefly describe the irregular ratio in the revision:

 *"Irregular ratio (IR) is a real number within the range [0, 0.5] to specify the random movement of the positions of successor-generation tetrahedra apexes to generate irregular particles. A larger IR makes the Koch-fractal geometry surfaces more irregular and asymmetrical."*

[Figure]

(a)        (b)        (c)        (d)

(a)–(d) The Koch-fractal particles of third generation with the same aspect ratio of 1.0. The irregular ratios of (a)–(d) are 0, 0.1, 0.2, and 0.3.

---

## Author Comment (AC2) · 19 Jan 2020

**Responses to the Short Comment by Patrick Stegmann (Manuscript # acp-2019-812)**

First of all, we would like to thank Dr. Stegmann for his valuable comments to the manuscript. In the revision, we have accommodated the suggested changes into consideration and revised the manuscript accordingly. All changes are highlighted in RED in the revision. In this point-to-point response, the reviewer's comments are copied as texts in BLACK, and our responses are followed in BLUE.

**General comment:**

This manuscript provides a good overview on the current state of mineral dust single scattering computations and addresses an important issue in the field of aerosol optics, namely the uncertainty in the particle refractive index.

Is there any specific justification for the Koch-fractal morphology compared to e.g. spheroids and super-ellipsoids? Is it more convenient or more accurate?
**Response:** In fact, there is no specific reason for the choice of the Koch-fractal morphology. Because this study focuses on the refractive index, and we simply consider a particular geometry that works. We considered the Koch-fractal particle because of its relatively accurate performance on representing dust optical properties (Liu et al., 2012; Jin et al., 2016). For example, Lin et al. (2018) revealed that neither spheroids nor super-ellipsoids can reproduce scattering matrix elements of red clay, whereas the Koch-fractal particles can (Jin et al., 2016). However, we think this study can also be performed using spheroids or super-ellipsoids, and we expect that they result in similar results. With respect to the scattering simulations, there is no difficulty for the PSTD and IGOM to consider Koch-fractal particles.

It would be interesting (but by no means necessary) to supplement this study with the derivatives of the phase matrices w.r.t. the refractive index.
**Response:** Thanks for the suggestion. We tried to present the derivatives, and the figure become difficult to read, as figure cover with each other and no clear trends are shown. Considering that the current Figures 3 and 4 are clear and can easily followed, we will not make the problem more complicated.

**Specific Notes:**

Page 1, line 10: Change "Dust" to "Mineral dust".
**Response:** Thanks. We have corrected it in the revision.

Page 1, line 14: "This study reveals the importance of the dust RI for the model development of dust optical properties"
**Response:** Thanks. We have corrected it in the revision.

Introduction in general: The IPCC has also identified aerosols as a major source of uncertainty in radiative forcing of the terrestrial climate. Maybe this can be added to the introduction with a suitable source to cite.
**Response:** Thanks, and we have include the following sentence in the revision:
"*According to the IPCC Fifth Assessment Report (IPCC, 2014), aerosol is still one of the largest sources of uncertainty in the total radiative forcing estimation.*"

P.2, l.4: "... mineral dust is widely distributed around the globe, ..."
**Response:** Corrected.

P.2, l.13: "For example, the measured phase functions of dust particles are distinctively different from the ...."
**Response:** Thanks. We have corrected it.

P.2, l.20: "a simplified but optically equivalent model is more convenient and easier to process"
**Response:** Thanks.

P.3, l.14f: "..., to which much less attention has been devoted during the model development."
**Response:** Thanks. We have corrected it in the revision.

P.5, l.15ff: Particle size distributions are a source of uncertainty as well. What is the reason for choosing the specific particle size distributions here?
**Response:** We consider simultaneous size distribution from the AGLSD, so there

should be less uncertainties from the size aspect. However, we completely agree with the reviewer, and think the size distribution can be a source of uncertainty, and this have been slightly investigated in previous studies, so we will not discuss it here. We just mentioned the potential further studies respect to uncertainties on particle size distribution.

P.6: The proposed method finds the optimal theoretical particle properties in terms of the phase matrix alone. What is the impact on the other properties, such as extinction coefficient, albedo etc.? Is a similarly good match achieved?

**Response:** Other optical properties are also sensitive to particle refractive index, whereas they are not measured simultaneously. We think this would be a great suggestion for future studies and instrument/observation design. We added the following sentence in the revision:

*"Last but not least, to better constrain either particle RI or geometry for dust optical property studies, more observations on dust microphysical and optical properties should be considered."*

---

## Author Comment (AC3) · 19 Jan 2020

**Responses to Anonymous Reviewer #2 (Manuscript # acp-2019-812)**

First of all, we would like to thank the anonymous reviewer for his/her thoughtful review and valuable comments to the manuscript. In the revision, we have accommodated all the suggested changes into consideration and revised the manuscript accordingly. All changes are highlighted in RED in the revision. In this point-to-point response, the reviewers' comments are copied as texts in BLACK, and our responses are followed in BLUE.

General comment:

This paper by Huang et al. reveals by model simulations the importance of the dust refractive indices (RIs) for the model development of its optical properties. They show that the scattering matrix elements of different kinds of dust particles can be reasonably reproduced by choosing appropriate RIs even using a fixed particle geometry and that a change in the RI can strongly affect the appropriate shape parameters to reproduce the measured dust phase matrix elements. The study indicates that the development of corresponding optical models can potentially be simplified by considering only variations over different RIs. The study should be a welcome addition to the literatures on modeling and measurements of dust optical properties and their radiative effects. The paper is well written in general, though the model used needs to be introduced more specifically and the parameters presented to be described more clearly. I would recommend the paper be published after minor revisions.

**Response:** Thanks the reviewer for the suggestions. The comments on the model, the presentations of parameters, and the result interpretations can significantly improve the quality of the manuscript, and make the paper more solid. The following presents our point-to-point responses as well as the revision for the manuscript.

**Minor/Technical issues:**

1. P2, L5: What implications of this study for better quantifying these two fundamental parameters? This issue may also need to be highlighted in the conclusions.

Response: Thanks for the constructive comments. In fact, single scattering albedo and

asymmetry factor are introduced as two examples of the various optical properties. This study mainly discuss the scattering matrix elements, so we omitted the two parameters in the revision. With better constrain on particle RI, the estimation on the SSA and asymmetry factor will definitely improve.

2. P2, L7: There are two literatures given in the References corresponding to Xu et al., 2017 here.

**Response**: We have used "Xu et al., 2017a" and "Xu et al., 2017b" to distinguish the two literatures, and the corresponding statements are corrected in the revision.

3. P2, L10: There are two literatures given in the References corresponding to Bi et al., 2018 here.

**Response**: The citations have been specified as "Bi et al., 2018a" and "Bi et al., 2018b" in the revision.

4. P3, L20: There are two literatures given in the References corresponding to Bi et al., 2011 here.

**Response**: Thanks for the careful review, and we have corrected the references accordingly. Sorry for the mistakes, and we have double checked the references.

5. P4, 111: Several types of dust particles?

Response: Thanks for the suggestion. We have corrected the phrase.

6. P6, Eq. (2): What are i and j stand for, respectively? Their ranges should be given in the equation. Should it be Pij in the denominator?

**Response**: As this study didn't consider  $d_{ij}$  other than  $d_{11}$ , so we simplified the discussion relative to the evaluation. In other words, Eq. (2) is unnecessary now, and we have deleted it.

7. P6, L6: Is this summation really used in the following sections? If so, an equation might be given here.

**Response**: We found that the optimal refractive index that gives the smallest  $d_{11}$  is normally consistent with that gives the smallest summation, so we consider only  $d_{11}$  in the manuscript and didn't directly use the summation here. To avoid confusion, we

have removed the corresponding discussions.

8. P7, L1-2: Do you mean element(s) or element ratio(s) here?

**Response**: Thanks for the suggestion. It is quite standard to present the scattering matrix elements besides  $P_{11}$  using their ratios to  $P_{11}$ , because the large variations can hardly be presented in the linear axis and the logarithmic axis cann't be used due to negative values. We think there will be little difference between element and elment ratio, so we try to keep text simple by using "elements" in the manuscript.

9. P7, L17: the literature for Nousiainen, 2014 is missing in the References.

**Response**: Thanks. The literature referred here should be "Nousiainen and Kandler, 2015", and we have corrected this in the revision.

10. P8, L1-2: Where is d11 shown in the figure? The phrase "element ratios" might be more suitable for P11/P11(30 degree), P12/P11, and P33/P11? Are all P11 in the denominator for 30 degree? Since these ratios are frequently used, their definitions (or meanings) need to be given in Sect. 2.

Response: Thanks for the constructive comments.

The  $d_{11}$  values are not directly shown in the manuscript, we just listed some of there as example. We give the various  $d_{11}$  with different RIs when the phase function of feldspar are discussed as examples (the smallest  $d_{11}$  together with the optimal RI are bold):

| Refractive index           | 1.4+10 -3 | 1.55+10 -3 | 1.55+10 -2 | 1.6+10 -3 | 1.8+10 -3 | 2.0+10 -3 |  |
|----------------------------|----------------------|-----------------------|-----------------------|----------------------|----------------------|----------------------|--|
| d 11 (Feldspar) | 13.56                | 0.51                  | 2.54                  | 0.87                 | 6.14                 | 10.12                |  |

(2). Thanks for the suggestion, we have added the following sentence in the text: "Noted that the phase functions will be presented by normalizing  $P_{11}(30^\circ)$  to 1, i.e., showing  $P_{11}(\theta) / P_{11}(30^\circ)$ , and the other nonzero scattering matrix elements are normalized with respect to  $P_{11}$ ."

11. P14, L2-3, L16-17, and L24-25; P16, L3-4; P17, L5-6 and L10-12; P18, L14-15 and L16; P19, L1-2 and L5-7: Are these literatures referred in the main-body text?

**Response**: Sorry for the mistakes. These literatures were referred in an early version of this manuscript, and we forgot to delete them for the submission. Some of those literatures are cited in the revision, and we have deleted these that are not referred. As mentioned above.

**12. P22, Fig. 3 and P23, Fig.4: Is P11 for 30 degree in all the Y-axis?**

**Response**: There is no problem with the labels, and we have double checked them. Because only the relative values of the phase function matters, it is normally presented with certain normalization. In this study, as the measurement phase function cannot be normalized by the standard integral value, we just present them by dividing the value at scattering angle of  $30^{\circ}$ , so there is  $P_{11}/P_{11}(30^{\circ})$  for the y-axis.

13. P24, Fig.5: Are measurements referred by black dots? Legends (or descriptions) of the plots need to be given.

**Response**: Thanks, the measurements are referred by black dots and the descriptions of the plots have been added to the corresponding caption in the revision.